# Prevalence of multidrug-resistant tuberculosis in East Africa: A systematic review and meta-analysis

**Kindu Alem Molla** [1]*, **Melese Abate Reta**[2], **Yonas Yimam Ayene**[1]

**1** Department of Biology, Faculty of Natural and Computational Sciences, Woldia University, Woldia, Ethiopia, **2** Department of Medical Laboratory Science, College of Health Sciences, Woldia University, Woldia, Ethiopia

* kindualem@wldu.edu.et

## Abstract

### Background

The rate of multidrug-resistant tuberculosis is increasing at an alarming rate throughout the world. It is becoming an emerging public health problem in East Africa. The prevalence of multidrug-resistant tuberculosis among pulmonary tuberculosis positive individuals in the region has not been thoroughly investigated.

### Aim

The aim of this systematic review and meta-analysis is to estimate the pooled prevalence of multidrug-resistant tuberculosis among newly diagnosed and previously treated pulmonary tuberculosis cases in East African countries.

### Methods

English published articles were systematically searched from six electronic databases: PubMed, EMBASE, Scopus, Science direct, Web of Science, and Google scholar. The pooled prevalence of multidrug-resistant tuberculosis and associated risk factors were calculated using Der Simonian and Laird's random Effects model. Funnel plot symmetry visualization confirmed by Egger's regression asymmetry test and Begg rank correlation methods was used to assess publication bias. A total of 16 articles published from 2007 to 2019 were included in this study. STATA 14 software was used for analysis.

### Results

Out of 1025 articles identified citations, a total of 16 articles were included in final meta-analysis. The pooled prevalence of multidrug-resistant tuberculosis among newly diagnosed tuberculosis cases and previously treated tuberculosis patients was 4% (95%CI = 2–5%) and 21% (95%CI: 14–28%), respectively. Living conditions, lifestyles (smoking, alcohol use, and drug abuse), previous medical history, diabetes history, and human immunodeficiency

**Data Availability Statement:** All relevant data are within the paper and its Supporting Information files.

**Funding:** The authors received no specific funding for this work

**Competing interests:** The authors have declared that no competing interests exist.

**Abbreviations:** CI, Confidence interval; HIV, Human immunodeficiency virus; MDR-TB, Multidrug-resistant tuberculosis; MTBC, *Mycobacterium tuberculosis* complex; OR, Odds ratio; PTB, Pulmonary tuberculosis; TB, Tuberculosis; WHO, World Health Organization; XDR-TB, Extensively drug resistance tuberculosis.

virus infection were risk factors contributing to the higher prevalence of multidrug-resistant tuberculosis in East Africa.

## Conclusion

The review found a significant prevalence of multidrug-resistant tuberculosis in the region. An early diagnosis of tuberculosis and rapid detection of drug-resistant *Mycobacterium tuberculosis* is a critical priority to identify patients who are not responding to the standard treatment and to avoid transmission of resistant strains. It is also very important to strengthen tuberculosis control and improve monitoring of chemotherapy.

## Introduction

Tuberculosis (TB) disease, the leading cause of mortality, caused by the *Mycobacterium tuberculosis* complex (MTBC), is killing about 2 million people each year globally and nearly half a million new cases of multidrug-resistant tuberculosis (MDR-TB) emerges every year [1]. It is ranked as the second leading cause of death from an infectious disease worldwide [2]. Countries such as Russia, China, and India alone accounted for 47% of the global burden. Different previous studies conducted in many countries have shown the MDR rate is much higher in previously treated TB patients than in new TB patients [3–5]. The highest TB mortality and morbidity occurs in middle and low-income countries like East African countries [6]. High mortality of TB disease was predominantly observed in the TB and human immunodeficiency virus (HIV) co-infected patients [7]. Currently, East Africa countries such as Ethiopia, Rwanda, Kenya, Tanzania, and Uganda are among the 30 high MDR-TB burden countries [8, 9]. According to Gobena et al. [10], Ethiopia is one of the high burden countries with regards to TB, TB/ HIV, and MDR-TB. According to the 2016 report, 89.7% of the global incident cases of MDR-TB were accounted for in 30 high MDR-TB burden countries. Among the 30 high burden TB countries ranked in the world, Kenya has the fifth-highest burden in Africa [7, 10].

MDR-TB is a major threat to global TB control strategies [11]. HIV co-infection MDR-TB did not get key attention until recently in East Africa, where the tuberculosis prevalence and risk factors are highest [12]. MDR-TB, despite being nearly 100% curable, cure rates remain below 100% in practice even in high-income settings, also it is one of the main public health problems that is most-frequently the cause of death among immune suppressed persons [13]. One-third of the world's population is infected with tuberculosis. The presence of the drug-resistant gene in *Mycobacterium tuberculosis* is the major challenge of controlling TB [14]. MDR (resistance to at least both to first-line anti-TB drugs such as isoniazid and rifampicin) TB occurred among 3.6% and 18% of new and previously treated TB cases, respectively (5.6% among all cases) [6]. Treatment of MDR-TB requires long time with a combination of second-line anti-TB drugs; most of them are less effective, has a considerable rate of adverse effects, and more expensive than first-line drugs [15].

Attention is not given to MDR-TB in East Africa and it has been ignored until recently in this region [16]. Even if there are some studies about the prevalence of MDR-TB in the region, only little is known about the associated risk factors to MDR-TB in East Africa [17, 18]. Little studies in the region showed that low socioeconomic status, history of anti-TB treatment, the presence of HIV co-infection, sex, history of diabetes mellitus, alcohol use, and malnutrition were the risk factors associated with the development of MDR-TB in the region [19, 20]. A

study done in Tanzania showed that risk factors associated with MDR-TB were previous history of treatment with first line anti-TB drugs (OR = 3.3, 95% CI 1.7–6.3), smoking (OR = 1.9, 95% CI 1.0–3.5), contact with TB case (OR = 2.7, 95% CI 1.4–5.1) and history of TB disease [21]. Similarly, a study done in Ethiopia revealed that a history of previous anti-TB treatment (AOR = 21; 95% CI: 17.8–28) and HIV co-infection (AOR = 3.1; 95%CI: 1.02–9.4) were found to be associated risk factors for MDR-TB [12]. There are a lot of published studies done on MDR-TB in different parts of East Africa. There were, however, insufficient reports on the pooled prevalence of MDR-TB in East Africa. This is might be due to poor laboratory facilities, outdated databases and poor surveillance mechanisms and reporting procedures. Therefore, we aimed to conduct this systematic review and meta-analysis to assess the pooled prevalence of MDR-TB in East Africa. Thus, this review would provide a current and comprehensive pooled prevalence of MDR-TB in East Africa.

## Methods

### Search strategy

A systematic literature search was conducted on six search engines, electronic bibliographic databases and libraries: PubMed, EMBASE, Scopus, Science direct, Web of Science, and Google scholar to retrieve potential articles published. English published articles conducted from the East African region from 2007 to 2019 were included.

The following search terms or phrases were used: "prevalence" OR "epidemiology" OR "magnitude" AND "multidrug-resistant tuberculosis" AND "East Africa". The search strings were applied using 'AND' and 'OR' Boolean operators. A total of 16 articles published in East Africa from 2007 to 2019 were included in this study.

### Inclusion criteria

For this systematic review and meta-analysis, we included cross-sectional, case-control and cohort studies reporting on the prevalence of MDR-TB among new pulmonary tuberculosis (PTB) cases and previously treated PTB cases for final analysis. All eligible studies published in English language conducted in East Africa from 2007 to 2019 were included.

### Exclusion criteria

Studies were excluded from the analysis for any of the following reasons: duplicated publication, articles available only in abstract form, case reports, anonymous reports, studies conducted outside of East Africa, studies reporting data regarding extra-pulmonary TB (EPTB), studies without adequate data, and studies published after September 2019 because of unfulfilled the inclusion criteria of MDR-TB among new PTB cases and previously treated PTB cases in East Africa.

### Data extraction

We used a standard data extraction format prepared in Microsoft Excel (Microsoft Corp., Redmond, WA, USA). The following data were extracted from included articles: Name of Author (s), year of publication, study period, study area, sample size, new cases of MDR-TB, and previously treated MDR-TB patients.

### Quality assessment

To evaluate the quality of all included studies, critical quality assessment checklist recommended by the Joanna Briggs Institute was used [22]. Two authors (KAM and MAR)

independently assessed the quality of included studies. A disagreement between the two authors was resolved through discussion. In the case of a persistent disagreement, a third author (YYA) was consulted to reach a consensus and to include articles in the final studies. The overall risk of bias assessment was rated based on the number of the high risk of bias per study: low ($\leq$ 2), moderate (3–4), and high ($\geq$ 5).

## Data analysis

For data analysis, the researcher used STATA 14 statistical software. The descriptions of original studies were assessed by using forest plots. To detect the presence of publication bias, funnel plot symmetry visualization followed by Egger's regression asymmetry test and Begg rank correlation methods were used. By using Der Simonian and Laird's random-effects model, point prevalence estimate of each study with a 95% confidence interval was used to estimate pooled prevalence.

# Results

## Search results

A total of 1025 articles were recovered from PubMed, EMBASE, Scopus, Science direct, Web of Science, and Google scholar [Fig 1]. Of these, 271 articles were excluded due to duplication, and, 665 articles were excluded after reviewing their title and abstracts. Then, the remaining 89 articles were identified for full text review. Of these, 16 articles were retained for final meta-analysis. We used PRISMA checklist for assessment of meta-analysis guideline compliance (S1 Table).

## Characteristics of included studies

East African countries such as 5 in Ethiopia (Admassu, 2011 [23], Mekonnen et al., 2015 [24], Brhane et al., 2017 [25], Biresaw et al., 2018 [26], Girum et al., 2018 [27]), 2 in Kenya (Kerubo et al., 2016 [28], Huerga et al., 2017 [29]), 1 in Rwanda (Umubyeyi et al., 2007 [30]), 3 in Sudan (Eldin et al., 2011. [31], Sabeel et al., 2017 [32], Eldirdery et al., 2017 [33]), 2 in Tanzania (Chonde et al., 2010 [34], Range et al., 2012 [35]) and 3 in Uganda (Lukoye et al., 2011 [36], Lukoye, 2013 [3], Okethwangu et al., 2019 [37]) were used in this systematic review and meta-analysis indicated in S2 Table. The references of all 16 studies included for the meta-analysis are provided in supporting information (S1 File).

## Publication bias and heterogeneity

To assess the presence of publication bias, the funnel plot symmetry visual inspection was used and the result showed the absence of publication bias [Figs 2 and 3]. The absence of publication bias was statistically confirmed by Egger's weighted regression test (bias coefficient (B) = 4.7, -2.18–5.46; P = 0.37) and (bias coefficient (B) = 3.4, (95%CI = -3.01–1.7; P = 0.54) for MDR-TB among newly diagnosed cases and among previously treated cases, respectively. The heterogeneity analysis for newly diagnosed and previously treated MDR-TB indicated that the occurrence of high variance among studies ($I^2$ = 94.7%, P$\leq$0.001) and ($I^2$ = 99.1%, P$\leq$0.001), respectively, which was statistically significant.

## Pooled prevalence estimate

In this meta-analysis, 11720 TB patients were included and 451 were new MDR-TB cases. Out of 9416 previously treated TB patients, 1842 were positive for MDR-TB provided in S2 Table. The pooled prevalence of MDR-TB among newly diagnosed TB cases was 4% [(95%CI: 2–5%);

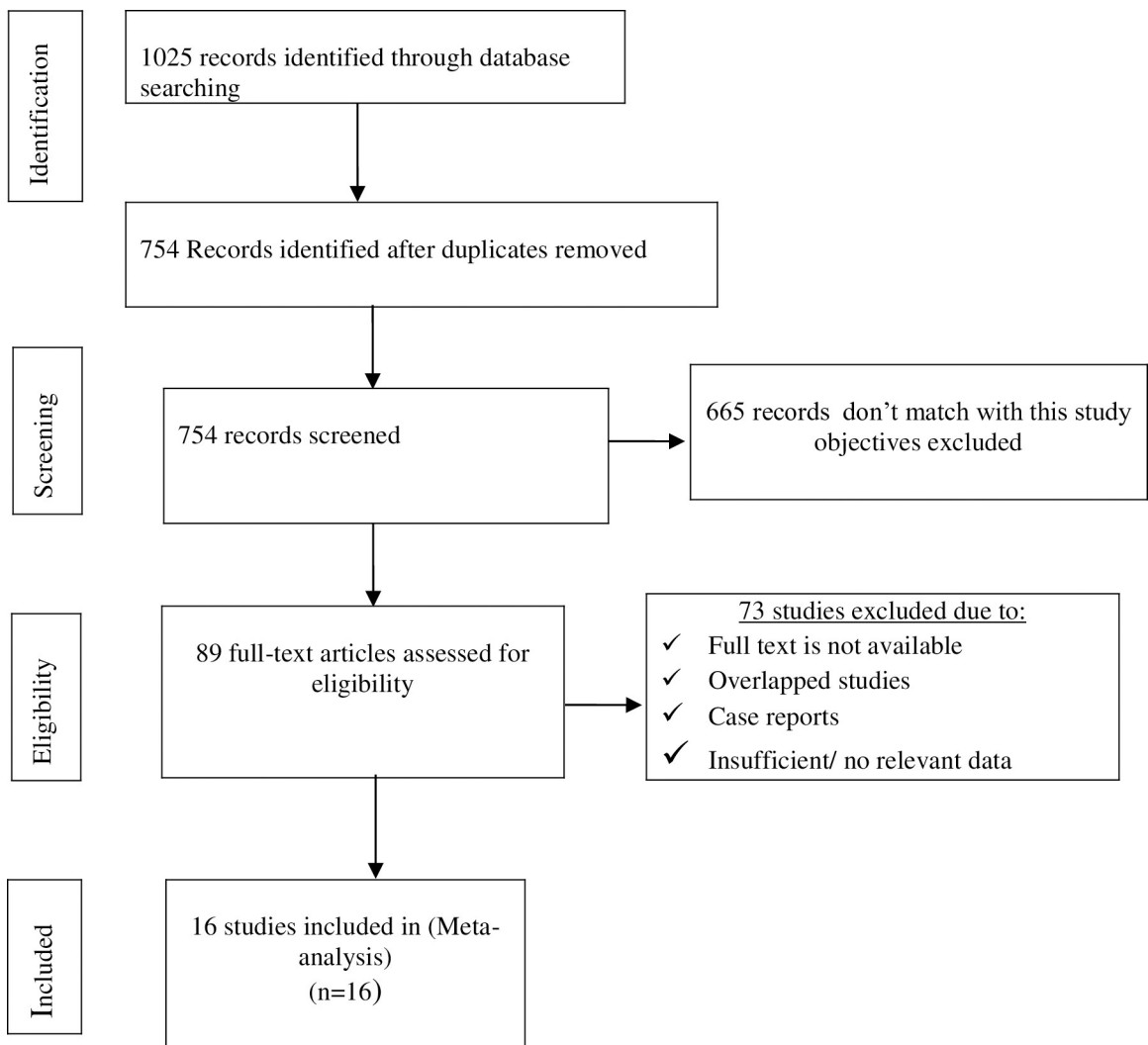

**Fig 1. Flow chart diagram showing study screening and selection procedure.**

$I^2$ = 94.7%, P≤0.001 [Fig 4]. Among newly diagnosed MDR-TB cases, the highest prevalence was reported to be 20% in Sudan by Sabeel et al. [32]. The overall prevalence of MDR-TB among previously treated TB cases was 21% (95%CI: 14–28%); $I^2$ = 99.1%, P≤0.001 [Fig 5]. The prevalence of MDR-TB among previously treated TB cases of each of the studies included in this systematic review ranges from 1% (95%CI: -0.00–1.00) by Range et al. [35] to 91% (95% CI: 81–101) by Okethwangu et al. [37].

## Discussion

Multidrug resistant tuberculosis is a major global health problem. TB continues to be a great health treat in developing countries and is compounded by the high burden of MDR-TB [38]. There is still a serious knowledge gap about MDR-TB in East African countries. Therefore, in order to find out the current circumstance of MDR-TB in the region, recent and evidence based studies are significantly required. Thus, this systematic review and meta-analysis addressed the prevalence of new MDR-TB and previously treated MDR-TB using 16 selected studies conducted from 2007 to 2019 in East Africa. In this meta-analysis, the prevalence of

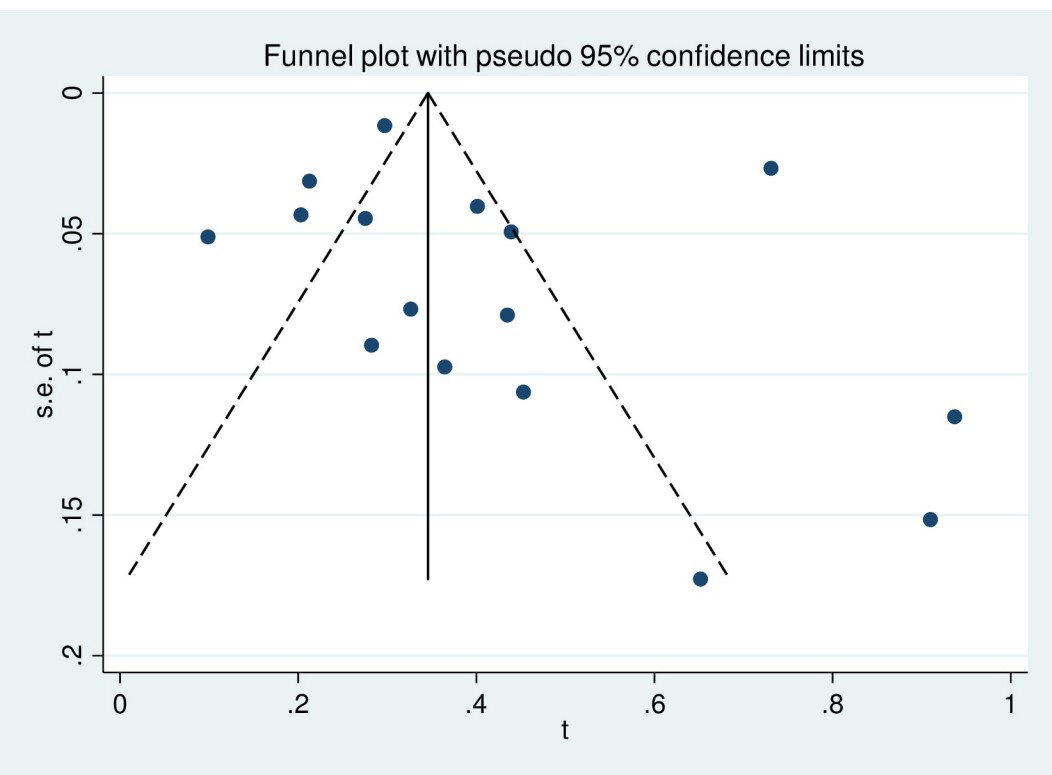

**Fig 2. Meta funnel plot for publication bias.** Prevalence estimate of MDR-TB among newly diagnosed TB cases in East Africa, 2007 to 2019. Abbreviation: t: arcsine transformed prevalence estimate of MDR-TB and se of t, standard error of t.

newly diagnosed MDR-TB ranged from 1% (95%CI: 0.0–2%) to 20% (95%CI: 11–29%) with a pooled prevalence of 4% (95%CI: 2–5%). The results of this meta-analysis showed that the pooled prevalence of newly diagnosed MDR-TB was somewhat higher than a previous meta-analysis report by Eshetie et al. [4] where 2% (95% CI 1–2%) of newly diagnosed TB patients have MDR-TB. According to the results of this meta-analysis, the prevalence of MDR-TB among previously treated cases ranged from 1% (95%CI:-0.00–1.00) to 91% (95%CI: 81–101) with a pooled prevalence of 21% (95%CI: 14–28%). Likewise, the same study done in Ethiopia [4] showed that the pooled prevalence of MDR-TB among previously treated cases was 15% (95% CI 12% - 17%) which was lower than the present study. In this study, the authors found that the level of MDR-TB were higher in East Africa than reported globally for both MDR among newly diagnosed TB patients and MDR among previously treated for TB patients. This study result showed that MDR-TB estimates as almost six times higher as compared to the global average reported by WHO for both MDR among newly diagnosed TB patients and MDR among previously treated for TB patients (21% vs 3.6%) [39]. According to this meta-analysis, the pooled prevalence of MDR-TB was higher than the pooled prevalence of MDR-TB with a previous meta-analysis report that was conducted in Ethiopia [27]. According to the WHO report of 2015, globally the prevalence of MDR-TB of new cases and previously treated cases of TB was 3.5% and 20.5%, respectively, while Sub-Saharan Africa countries contributing the highest proportion and these levels in recent years have remained unchanged [40, 41]. The main reasons behind the appearance of MDR-TB globally are multi-factor and are related to living conditions [42], life style [21], previous medical history [43, 44], diabetes history [45, 46], HIV infection [47], and education level [43].

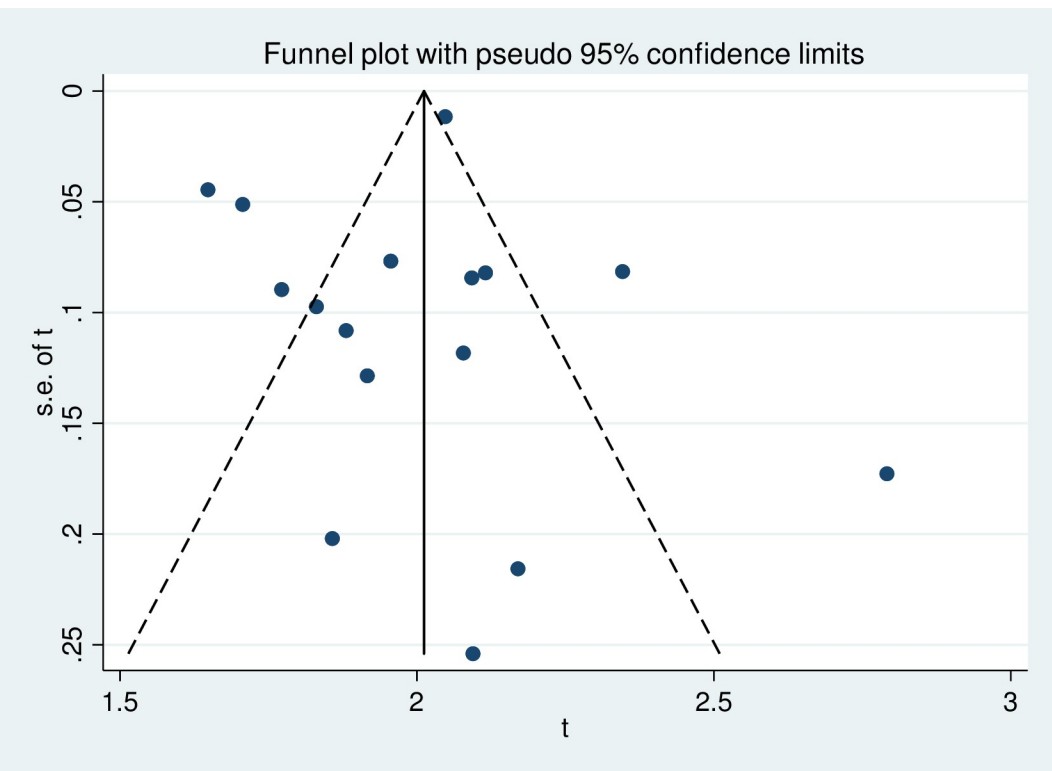

**Fig 3. Meta funnel plot for publication bias.** Prevalence estimate of MDR-TB among previously treated TB cases in East Africa, 2007 to 2019. Abbreviation: t: arcsine transformed prevalence estimate of MDR-TB and se of t, standard error of t.

Poor and crowded living conditions of a family might make easy the spread of TB. Thus, these lower socioeconomic family groups should get the highest concern for MDR-TB prevention efforts [48]. Patients that live in a household with more than one room were five times at lower risk of having MDR-TB than those living in a household with only one room [12]. This might be as a result of high risk of getting resistant strains from infected people in crowded places. Different lifestyles such as alcohol abuse, smoking, drugs use, etc. are the major risk factors associated with the development of MDR-TB. Cigarette smokers were more likely to be infected with MDR-TB infection and have less chance to be cured. Male cigarette smokers were more likely to be infected with MDR-TB and susceptible compared than females. This might be due to that men are predominantly drink alcohol, smoke and consume the drug compared to women [10, 49–51].

The longer exposure of a patient to anti-tuberculosis drugs was also associated with the development of MDR-TB. Patients under multiple situation of anti-tuberculosis treatment might create greater antibiotic resistance with the consequent development of MDR-TB and extensively drug resistance tuberculosis (XDR-TB) cases. Previous TB disease and chemotherapy are the most important risk factors associated with MDR-TB [43, 44, 52]. The global diabetes mellitus (DM) epidemic creates a serious bottleneck to the TB control program [53]. Individuals with diabetes, as compared to non-diabetic controls, were two-to three-folds more likely to develop TB. Weakened immunity in diabetic patients is thought to contribute to the development of latent TB infection to active cases [45, 46].

A study done in Ethiopia revealed that HIV infection was identified as a significant factor associated for MDR-TB [26]. Moreover, individuals living with HIV might also be more likely

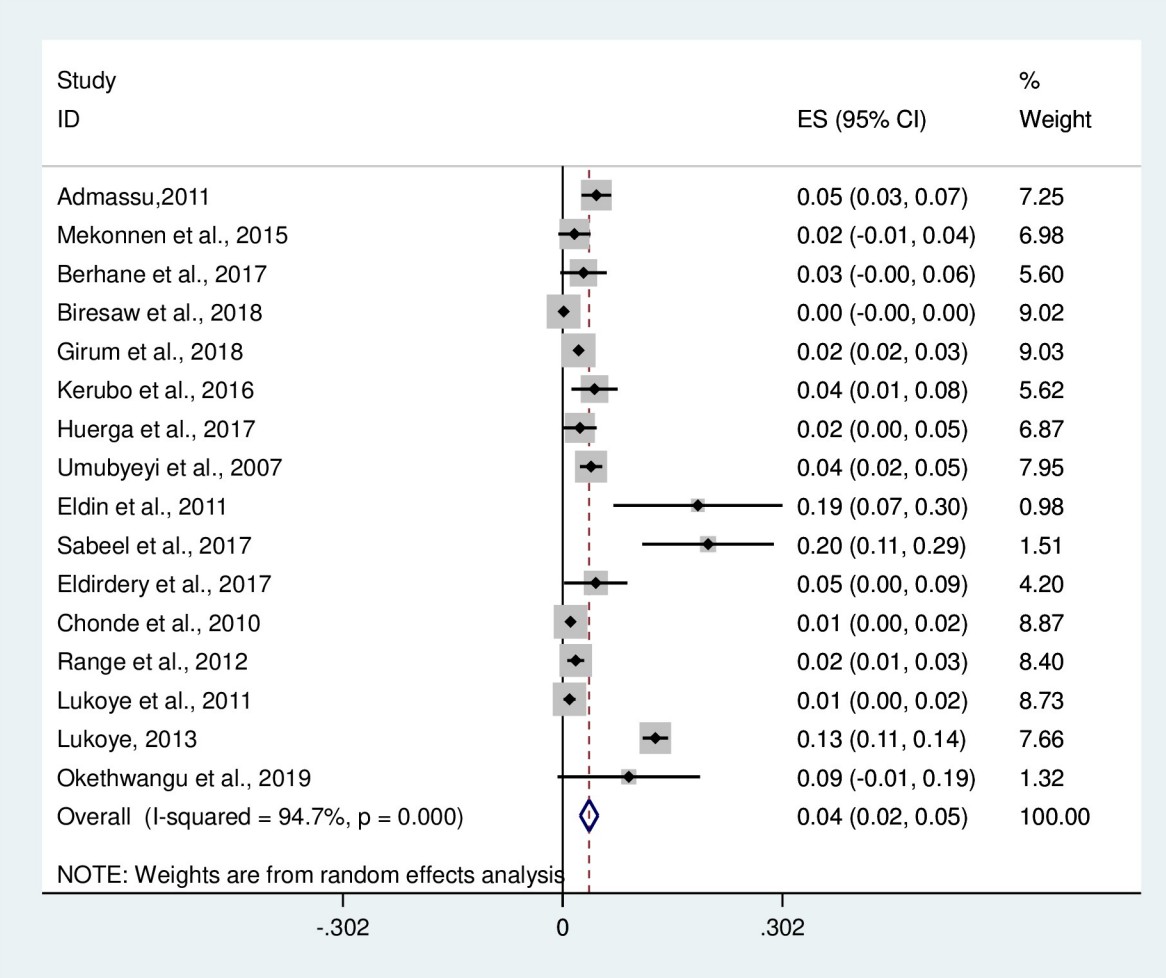

**Fig 4. Forest plot showing the prevalence of MDR-TB among new diagnosed TB cases in East Africa, 2007 to 2019.** Abbreviation: ES: Effect size; CI: Confidence interval.

to be exposed to MDR-TB because of longer hospitalization in settings with low infection control. Drug malabsorption in HIV infected people can also lead to drug resistance and has been shown to result in treatment failure [54]. In one study conducted by Al-Darraji et al. [55] revealed that there was 20% higher occurrence of MDR-TB among HIV-positive persons than those HIV-negative persons. Patients with low education level have been associated with the development of MDR-TB [43]. A study conducted by Ronaidi et al. [56] showed that the number of MDR-TB infected patients is significantly higher among the lower education group compared to the higher education group. It has been reported that the higher prevalence of PTB with unsuccessful treatment outcome was observed in patients who had a lower education level [57]. However, studies conducted in Ethiopia and Sudan showed that educational status was not significantly associated with MDR-TB [58–60].

## Limitations

This review had some limitations. Out of 11 East Africa countries, only 6 countries had done studies that fulfilled the inclusion criteria. This review has been summarized the findings of 16

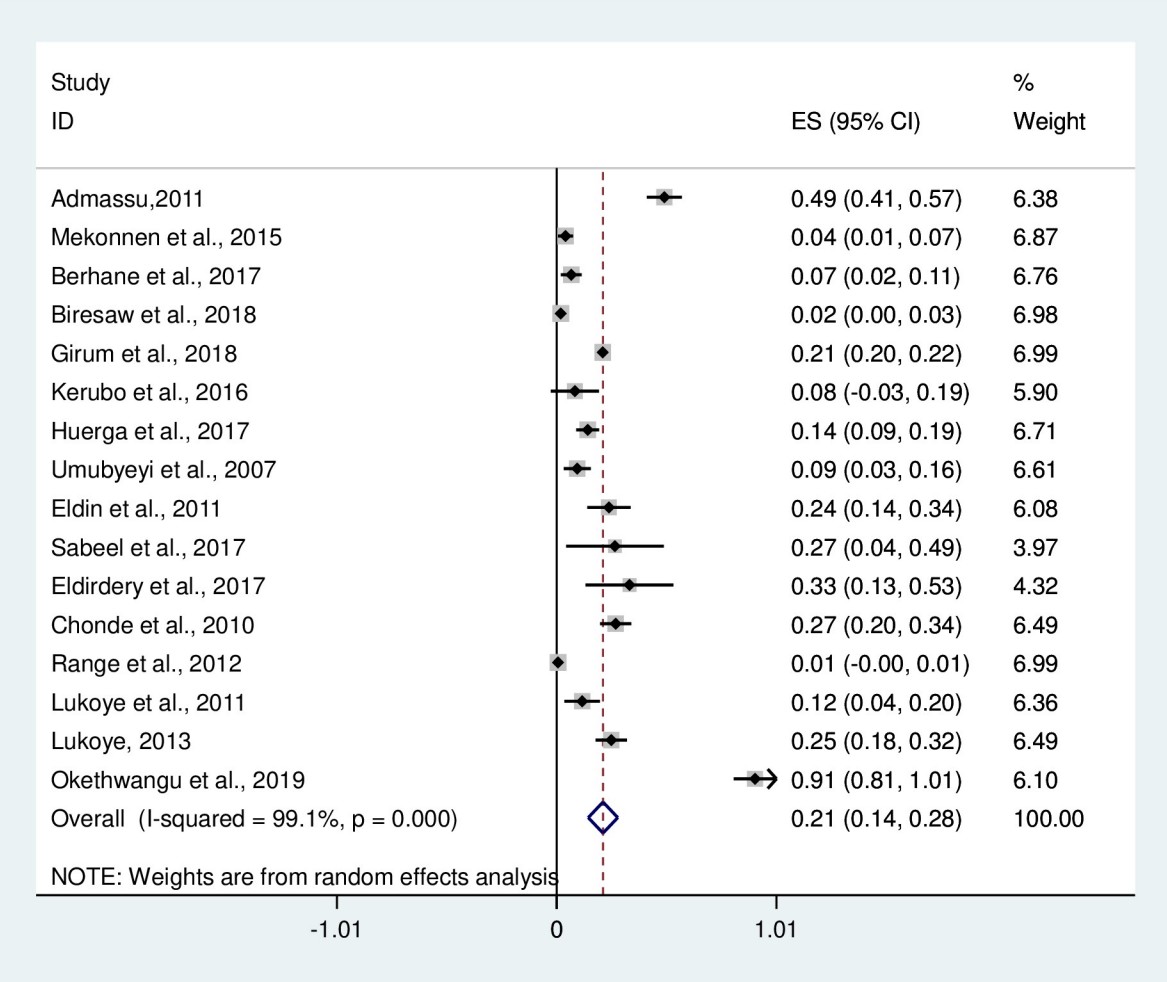

**Fig 5. Forest plot showing the prevalence of MDR-TB among previously treated TB cases in East Africa, 2007 to 2019.** Abbreviation: ES: Effect size; CI: Confidence interval.

published articles about the prevalence of MDR-TB in East Africa. The findings of this review are limited because of a small number of study countries, small number of published articles, and small sample size of study samples conducted on the prevalence of MDR-TB. However, the findings of this review give reliable results on MDR-TB in East Africa.

## Conclusion and recommendation

In conclusion, this systematic review and meta-analysis indicated a high proportion of MDR-TB in PTB patients in East Africa. Most of the studies in this review showed that the pooled prevalence of MDR-TB in TB positive individuals is higher by far in the study area than results reported in previous studies. In the present study, the pooled prevalence of MDR-TB among previously treated patients is higher than the pooled prevalence of new cases of MDR-TB. The majority of the studies in this review indicated that, living conditions, life-styles (smoking, alcohol use, and drug abuse), previous medical history, diabetes history and HIV infection risk factors contribute to higher prevalence of MDR-TB in East Africa. A special attention should be given in the TB control program on improving the patient's adherence to

anti-TB drugs, and health promotion activities about TB. It is also mandatory to create aware-ness about the mode of transmission of TB for people living in a household with only one room. Effective measures also need to be implemented to promote early diagnosis of MDR-TB and treatment before complication.

## Supporting information

**S1 Table. PRISMA checklist of systematic review.**
(DOCX)

**S2 Table. Characteristics of studies included in this meta-analysis.**
(DOCX)

**S1 File. References of studies in meta-analysis.**
(DOCX)

## Acknowledgments

The authors of this systematic review and meta-analysis thank all corresponding authors who had investigated and provide information about the prevalence of MDR-TB, and associated risk factors.

## Author Contributions

**Conceptualization:** Kindu Alem Molla.

**Data curation:** Kindu Alem Molla, Yonas Yimam Ayene.

**Formal analysis:** Yonas Yimam Ayene.

**Methodology:** Kindu Alem Molla, Melese Abate Reta.

**Software:** Yonas Yimam Ayene.

**Validation:** Kindu Alem Molla, Melese Abate Reta.

**Writing – original draft:** Kindu Alem Molla.

**Writing – review & editing:** Kindu Alem Molla, Melese Abate Reta, Yonas Yimam Ayene.

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
