## [Decision Letter · Decision Letter 0]

14 Mar 2022

PONE-D-22-00221Prevalence of multidrug-resistant tuberculosis in East Africa: A systematic review and meta-analysisPLOS ONE

Dear Dr. Molla,

Thank you for submitting your manuscript to PLOS ONE. After careful consideration, we feel that it has merit but does not fully meet PLOS ONE’s publication criteria as it currently stands. Therefore, we invite you to submit a revised version of the manuscript that addresses the points raised during the review process.

We look forward to receiving your revised manuscript.

Kind regards,

Mohmmad Younus Wani, Ph.D

Academic Editor

PLOS ONE

Journal Requirements:

Reviewers' comments:

Reviewer's Responses to Questions

**Comments to the Author**

1. Is the manuscript technically sound, and do the data support the conclusions?

Reviewer #1: Yes

Reviewer #2: Yes

Reviewer #3: No

Reviewer #4: Yes

2. Has the statistical analysis been performed appropriately and rigorously? 

Reviewer #1: Yes

Reviewer #2: I Don't Know

Reviewer #3: Yes

Reviewer #4: Yes

3. Have the authors made all data underlying the findings in their manuscript fully available?

Reviewer #1: Yes

Reviewer #2: Yes

Reviewer #3: No

Reviewer #4: Yes

4. Is the manuscript presented in an intelligible fashion and written in standard English?

Reviewer #1: Yes

Reviewer #2: No

Reviewer #3: No

Reviewer #4: Yes

5. Review Comments to the Author

Reviewer #1: The study has been written in a comprehensive and scientific language and followed the suitable methodology and statistical testing. Only to revise the conclusion and stick to the aim. Generally, no major flaws.

Reviewer #2: 1. The paper needs revision in view of spelling, English grammar.

2. Plagiarism was found which was more than expected

3. Key word, prevalence ‘P’ letter was written in a lower letter

4. Since the topic of this manuscript focuses on Prevalence MDR in East African countries, the study area was not included countries fairly in the study, e.g. Djibouti, Somalia, Eritrea, and the rest and also the included countries not equally or fairly distributed the study area according to table 1 page 8 and line 202 e. g Ethiopia 5 times whereas Rwanda is one time and so on

Reviewer #3: The manuscript deals with a systematic rveiview cum metaanlysis of the prevalence of MDRTB in East Africa. The english language in the entire script needs serious attention and is not as per the standards. The data sets for example Range et al which has been included in the analysis has the data collected during 2001 and 2002 period and the study states that the data is from 2007, so this has to be carefuly checked and included or excluded in the study. As this a prevalence stduy and the concern exists about collecting and pooling data from different time points and that too it ranges in years, the authors has to justify this if they are going to discuss about prevalence of the disease. The discussion is very wierd in certain aspects, for instance in lines 293 294 the authors state that "The longer exposure of a patient to anti-tuberculosis drugs was also associated with the development of MDR-TB" is a statement very difficult to accept and the subsequent sentences could not be understood. Most of the data has been rounded off and the data when cross checked are very dissimilar and the rounding off should have been standard and atleased quoted. The data and the analysis for the rsikfactors identified/stated are completely lacking and are not supported by any data ands could not be found in the manuscript. Hence the study lacks any inovative findings from the metaanalysis and the conclusion is all that could be found in text books.

Reviewer #4: My comments of minor nature are as under:-

Language issues are in plenty. For eg. in line 273-276, authors state that their MDR-TB estimates were six times higher as compared to the global average reported by WHO. I am sure the authors do not mean this.

Similarly, some of the terms are used loosely like new MDRTB, newly diagnosed MDR TB, previously diagnosed MDR TB, etc. What the authors mean is MDR among newly diagnosed TB patients and MDR among previously treated for TB patients. They should be consistent and careful in using terms.

Line 70: cure rate among MDR TB remains below 100%: Need to reframe

There are language issues in line 181-182.

Some of the references do not seem to be pertinent; for example, ref 1 &2.

Line 86-87: history of TB infection be replaced by TB disease

6. PLOS authors have the option to publish the peer review history of their article (what does this mean?). If published, this will include your full peer review and any attached files.

Reviewer #1: **Yes: **Layth Al-Salihi

Reviewer #2: **Yes: **Sebsib Neway Wolderufael

Reviewer #3: No

Reviewer #4: No

---

## [Author Response · Author response to Decision Letter 0]

7 May 2022

We the authors would like to thank the editor and reviewers for their constructive comments. In the revised submission we have included a marked-up copy of our manuscript that highlights changes made to the original version, Revised Manuscript with Track Changes, an unmarked version of our revised paper without tracked changes, and responses to editor and reviewers.

---

## [Decision Letter · Decision Letter 1]

8 Jun 2022

Prevalence of multidrug-resistant tuberculosis in East Africa: A systematic review and meta-analysis

PONE-D-22-00221R1

Dear Dr. Molla,

We’re pleased to inform you that your manuscript has been judged scientifically suitable for publication and will be formally accepted for publication once it meets all outstanding technical requirements.

Kind regards,

Mohmmad Younus Wani, Ph.D

Academic Editor

PLOS ONE

Additional Editor Comments (optional):

Reviewers' comments:

Reviewer's Responses to Questions

**Comments to the Author**

1. If the authors have adequately addressed your comments raised in a previous round of review and you feel that this manuscript is now acceptable for publication, you may indicate that here to bypass the “Comments to the Author” section, enter your conflict of interest statement in the “Confidential to Editor” section, and submit your "Accept" recommendation.

The authors have addressed the comments raised by the reviewers and the manuscript is now ready for publication.

Reviewer #1: All comments have been addressed

Reviewer #3: (No Response)

2. Is the manuscript technically sound, and do the data support the conclusions?

Reviewer #1: Yes

Reviewer #3: Partly

3. Has the statistical analysis been performed appropriately and rigorously? 

Reviewer #1: Yes

Reviewer #3: Yes

4. Have the authors made all data underlying the findings in their manuscript fully available?

Reviewer #1: Yes

Reviewer #3: Yes

5. Is the manuscript presented in an intelligible fashion and written in standard English?

Reviewer #1: Yes

6. Review Comments to the Author

Reviewer #1: The authors succeeded in expressing their findings and responded well to reviewers' comments. The context of the article matched regular scientific methods and the conclusions were coherent with the results and aims.

7. PLOS authors have the option to publish the peer review history of their article (what does this mean?). If published, this will include your full peer review and any attached files.

Reviewer #1: **Yes: **Layth Al-Salihi

Reviewer #3: **Yes: **Azger Dusthackeer

---

## [Editor Report · Acceptance letter]

14 Jun 2022

PONE-D-22-00221R1 

Prevalence of multidrug-resistant tuberculosis in East Africa: A systematic review and meta-analysis 

Dear Dr. Molla:

I'm pleased to inform you that your manuscript has been deemed suitable for publication in PLOS ONE. Congratulations! Your manuscript is now with our production department. 

Kind regards, 

on behalf of

Dr. Mohmmad Younus Wani 

Academic Editor

PLOS ONE